# Elusive Users: The Presence of Physically Disabled Users within Architectural Design Processes

**Marcus Tang Merit** [1,*], **Masashi Kajita** [2] **and Jonna Majgaard Krarup** [1]

1   Institute of Architecture, Urbanism and Landscape, Royal Danish Academy of Architecture,
    Design and Conservation, 1435 Copenhagen, Denmark
2   Institute of Architecture and Design, Royal Danish Academy of Architecture, Design and Conservation,
    1435 Copenhagen, Denmark
*   Correspondence: mtan@kglakademi.dk

**Abstract:** This paper is based on 8 months of sociological participatory fieldwork at the office of Gottlieb Paludan Architects, following the design process of a new concourse area for Ny Ellebjerg Station in Copenhagen, Denmark. The study aims to trace what presence users with physical disabilities possessed during a design process in which they were not physically present or explicitly involved. The study bases its findings on the visual material produced during the design process by the employees of Gottlieb Paludan Architects as well as the thoughts and discussions of practitioners. Drawing on actor-network theory, the study describes and analyses these human and non-human actors as they constitute and contribute to the design process. The study finds that users with physical disabilities were present within the design process through an implicit *generalized presence* and an explicit *required presence*. *Generalized presence* refers to those instances where the architectural qualities that were strived for in the project implicitly aligned with the needs of users with physical disabilities. *Required presence* refers to those instances during the design process where accessibility demands from client guidelines or building regulations played an important role.

**Keywords:** design processes; disability; user presence; actor-network theory; interdisciplinarity; sociology; Copenhagen

## 1. Introduction

There exists a wealth of literature pertaining to how the cities and urban spaces we share can be designed to be socially sustainable and just. To name just a few examples of such "cities", we have the feminist city [1], the restorative city [2], the open city [3], and the human -sized city [4]. This list could go on. One thing these and other such "cities" have in common is an attempt to influence designers and architects on how to best shape the urban habitats of human beings. However, sustainability, be it social or otherwise, and other societal issues are not effortlessly translated into everyday architectural practice [5], despite the effort that goes into researching and disseminating the ideal city. This means there exists a pragmatic gap in our understanding of how architects can best address varying societal challenges in a professional reality of multiple competing and intersecting interests. The previous literature has worked towards filling this gap by exploring the dependencies and relations of architecture [6], the architectural design process itself [7], and how to involve users' knowledge in design processes [8,9]. In relation to this, previous research has emphasized the creative potential of inclusionary design practices when working with vulnerable groups [10,11]. These studies are particularly interested in what can and does happen when architects collaborate with vulnerable groups. However, in the architectural project that forms the case of this study and, arguably, in many other architectural projects in Denmark and other countries, such direct collaboration is not the norm.

This article presents a participatory field study in Gottlieb Paludan Architects' (GPA) design process of the concourse area of Ny Ellebjerg Station (NES) in Copenhagen. More

specifically, it delivers an empirical exploration of what presence users with physical disabilities have in the day-to-day work of practicing architects. Here, it is important to mention that users with physical disabilities never had any direct presence through user involvement or similar processes during the design process. Exploring why and how such a group's presence still matters to the design process is a central point to this paper. In Denmark, 25% of the population suffers from mild (16%) or severe types of physical disabilities (9%), which makes them significantly less able to use Danish public transportation [12]. This should make the accessibility and inclusivity of public transport an important social sustainability agenda; however, recent research indicates that this is not currently the case. When questioned, few practicing Danish landscape architects tie social sustainability to accessibility before being explicitly questioned if there is a link between the two [13]. Furthermore, by drawing attention to users with physical disabilities, this article emphasizes a particular group of users whose spatial experiences could be considered phenomenologically distinct from the spatial experiences of "able-bodied" architects and users [10,14]. Such a distinction has previously led disability theorists to criticize design practices within architecture and other professions as "normate", problematizing the lack of insight gained from users with physical disabilities and the lack of design taking into consideration the body and neurodiversity of human beings [15,16]. However, previous normate critiques of architectural design processes lack empirical purchase in the daily realities of practicing architects' attempts at mediating numerable opposing interests into the design of future spaces.

Therefore, the empirical case and analytical focus of this paper is to show an example of how architects work with societal challenges with which they might have no intimate knowledge but are nonetheless expected to help solve. Thus, this provides a window of opportunity for studying the everyday work of architects to understand what type of presence users with physical disabilities have within an architectural design process and how this presence influences the decisions that architects make about their designs. Therefore, this article asks the question: How are users with physical disabilities present in architectural design processes through visual devices?

By devices, this study means the myriad forms of visual, textual, and physical materials produced and distributed during architectural design processes. This is a broad category that, among other things, refers to the numerous types of drawings, digital files, physical models, material samples, site photographs, reports, 3D renderings, and virtual reality models present during architectural design processes. Previous social scientific research has extensively emphasized the importance and agency of these devices through their constant circulation within architectural design processes [17–21]. It is, therefore, the argument of this paper, based on empirical experiences and previous research, that devices form an unavoidable element in the architectural design process and should figure as part of this study's central research question. Previous primarily social scientific research, despite its necessarily cross-disciplinary emphasis in studying a different profession's everyday work routines, has, to a large degree, ignored what might be learned by introducing the visually founded methods of architectural spatial analysis to the world of academic social scientific analysis [22]. The limited research that does exist by social scientists attempting to breach this methodological and disciplinary gap nonetheless proves the fruitfulness of challenging the analytical conventions of both social science and architecture [23,24].

By drawing equal attention to the devices being produced by architects and to the conversations architects have about and around devices, this study aims to provide findings that can be understood by both architects and social scientists [25]. In order to achieve this equality, this study draws upon actor-network theory (ANT), which provides a methodological framework for symmetrically describing and engaging with both human and non-human actors within the design process [26]. By providing a symmetrical description of both the architects and their devices, this study aspired to dispel the elusiveness that can exist when social scientists and architects try to agree on how users might be present within design processes.

## 2. Data Collection and Ny Ellebjerg Station's Future Concourse

As a train station, NES services both national and regional train lines from three different platforms. Two of its platforms are positioned at street level and run parallel to one another. The last platform is raised above ground, spanning across the tracks at ground level (see the top-left picture in Figure 1). To change between platforms requires several hundred metres of walking and the navigation of multiple flights of stairs or elevators. Figure 1 shows a set of photos taken from different access points to the station and emphasizes the lack of spatial coherency NES currently suffers from. The new concourse of NES is planned to open in 2024, simultaneously with a new subterranean metro station currently under construction at the site. Furthermore, there are plans to expand NES with an additional train platform at street level to service national and European train lines. With these additions to NES, and the significant urban development taking place on all sides of the station, NES is projected to become one of Denmark's busiest stations by 2040. As such, NES will serve as an important station for commuters to change between national, regional, and local public transit lines in Copenhagen. The key challenge for the concourse project is, therefore, to design a coherent space between the many means of public transportation connected to NES. As the concourse, in the future, needs to connect to various urban development areas placed on both sides of the rail tracks and because of the spatial constraints of the urban area, the concourse is planned to be constructed 5 m below street level. This also allows for the concourse to easily connect to the new metro station, something of particular importance to the Copenhagen Metro, which serves as the primary client for the concourse project.

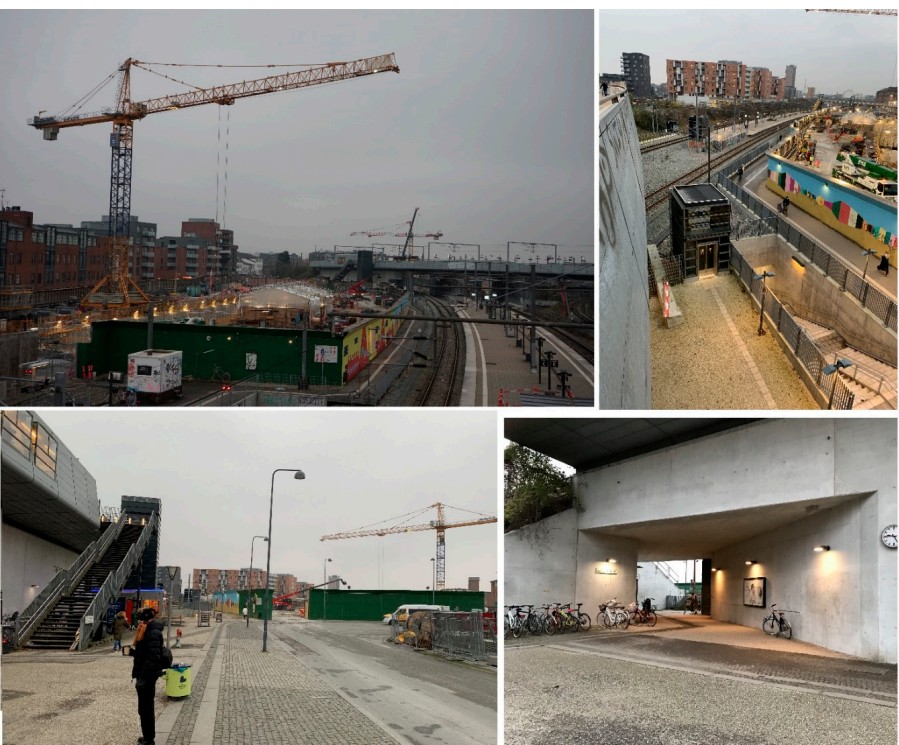

**Figure 1.** Images of the NES site in February 2021.

The development of NES has undergone several iterations, beginning as early 2016 with an earlier study conducted by GPA of how a future station might look. The scale of the urban development taking place around NES and the future plans for the station itself make this project too complicated to describe in its entirety in this study. It is nonetheless important to grasp that the development of the NES concourse is just one of many phases of future development planned, with varying degrees of certainty, for the station. Some of these plans involve the expansion of the station to include two new entrances and a

potential over-site development, to pay for some of the costs related to the subterranean metro station. Contractually, these plans were kept out of the job that GPA was hired to do by the Copenhagen Metro. Naturally, the future intentions for the surrounding area nonetheless did influence the discussions and decisions made throughout the design process at GPA. It is furthermore important to explain that the Copenhagen Metro is acting as the sole client for the concourse project and that the overall project management is contracted to a large external architectural, engineering, and consultancy company. Here, the Copenhagen Metro also represents the interests of other public organizations such as the municipality of Copenhagen and the Danish State Railways, with whom they will share the finished concourse space. As a company, GPA has extensive experience with these external actors from previous projects. In addition, some of the architects working at GPA are former employees of the Copenhagen Metro. This resulted in a friendly and at times informal collaborative process throughout most of the project.

The study into the concourse's architectural design process is based on 8 months of participatory fieldwork at Gottlieb Paludan Architects (GPA), starting September 2020 and ending May 2021. During this time, fieldwork was conducted through a full-time physical presence at the office of GPA, apart from the roughly 3 months when COVID-19 caused the team involved with NES to work from home, in which case the fieldwork was limited to the online meetings taking place as part of the design process. This study follows the project during three "stages", which are referred to as conceptual design (primarily focused around ideation), preliminary design (primarily focused around the geometry of stairs, escalators, elevators, and other elements), and detailed design (primarily focused on lighting and materials). The data collected from this fieldwork are comprised of some 34 pages of field notes corresponding to 35 audio-recorded meetings and various non-recorded meetings that took place around the office. A catalogue of devices was also produced, resulting in 47 entries by devices, with corresponding descriptions and notes. The field notes, recordings, and device catalogue were all time stamped and continuously linked to each other to form a coherent empirical material. To gain unmediated access to the design process while work was taking place [27] this study employed participatory ethnographic fieldwork. This method has the significant methodological benefit of overcoming potential dissimilarities between rhetoric and action [28], while also allowing researchers to make note of how such dissimilarities manifest themselves. In accordance with this study's interdisciplinary intentions, visual and textual empirical data are granted equal attention in order to provide a valid representation of the architectural design practice. Methodologically and analytically, this means the empirical data, in the form of the visual and physical materials used throughout the architectural design process (i.e., devices), are granted equal epistemological value to comments and conversations (i.e., field notes and audio recordings).

## 3. Devices, Users' Presence, and Actor-Network Theory

### 3.1. Why Devices Matter

Much research on design processes has traditionally privileged accounts that lend primacy to either social, cognitive, technical, historical, or organizational explanations [29]. This has made theorists comment on the presence of agential cuts in research, referring to the theoretically imposed limitations on who or what is granted agency [30]. Cuts such as these have led to discussions about whether the object or subject should take precedence in their explanative capabilities, which in turn would challenge this study's argument that devices and architects should be granted equal attention in the study of design processes. Pragmatist theorists have pointed out that such discussions mistake the premise of agency by asking with whom agency lies, rather than acknowledging the equal presence of both human and non-human actors. Their point is that it is the researcher, not the empirical world, who methodologically and theoretically cuts what is engaged with as an actor [31].

This emphasizes the importance of making it transparent how devices matter and how devices come to matter, both from a theoretical perspective and as an explicit analytical

goal for this study. To specify this, Noortje Marres, albeit in the context of politics and publics, states that we should examine "how material entities become invested with specific capacities ( . . . ) in particular settings and at certain times" [32]. This is not an argument for a relativistic understanding, in which devices are merely objects through which subjects manifest their agency. Indeed, Marres states that it is not enough to see how materials enter into situations, as we should focus our attentions on "how the material form of participation is actively accomplished with the aid of devices" [32]. Once constructed, devices become imbricated, as layers of meaning are overlapped unto them over time by other actors, with this process itself being tied to what agency the device affords other actors through its physical and visual characteristics [33]. While the affordance of a device can be summarized as the breadth of actions made available to actors by the device [34], it is implausible to describe the wealth of empirically observed actions tied to it [35]. This directs the attention of this study to the continual imbrication of devices, as these are constructed by architects and subsequently enter into the continually expanding network of actors surrounding any architectural project [36]. Architects shape and are shaped by the devices introduced throughout design processes.

### 3.2. Tracking Actors in the Design Process

Methodologically, this requires the introduction of a framework through which to track the construction and usage of devices in architectural design processes, as these take on myriad roles in everyday work, while simultaneously not forgetting about the architects amid these processes. In other words, we must follow the actors, whoever or whatever they might be, as these constitute and maintain the network that is the architectural design process and subject of this study [37]. When devices are referenced throughout this study, it is, therefore, not important to think of this visual material in terms of a singular image or several images merged into a PDF presentation; what matters is whether the image or the presentation as a device enacts influence upon other actors in the network. This is an empirical question to be answered through data gathering and subsequent analysis, not a question about making an agential cut to limit what type of visual material constitutes devices worth mentioning. What matters is sticking to the empirical data presented by following the design process, as they are constituted by the attachments and interactions between architects and devices [38]. Therefore, the design process forms a heterogeneous network of entities that are deigned important precisely because, and only insofar as, they enact influence on other entities in the design process [39].

While this principle of symmetry applies a highly empiricist notion of the studied actor-network, it is wrong to assume this means that this study or ANT in general can assume an atheoretical stance, or, indeed, that ANT provides a platform for unbiased observations [40]. Here, it is important to recall that due to theoretical, pragmatic, or thematic constraints imposed on studies by researchers or external actors, agential cuts, and, as such, limits to how far researchers can trace the network, are always present. To this end, the research question of this study provides an example of such agential cuts, as this study is explicitly interested in tracing the presence of physically disabled users. This directly influences which parts of the network are described here and which are omitted. However, following the principles of actor-network theory, it is important to leave it up to the actors to describe how the design process unfolds within the "scale" of this research question as "( . . . ) scale is what actors achieve by scaling, spacing, and contextualizing each other through the transportation in some specific vehicles of some specific traces" [37]. Therefore, if we are to trust Albena Yaneva when they state that "reality is exported from architectural practices not in the form of big theories, visions, manifestos, but rather in the shape of scale models, renderings, videos and drawings" [36], this study needs to represent the studied network through the visual means and language of the network. Only then can the study answer its research question truthfully, as it produces well-constructed arguments in a language that holds sway both from the perspective of architects and researchers [26].

*3.3. Presence of Physically Disabled Users*

This leads to the central concern of this study, and the validity of any conclusions that are drawn here. How, after all, can we discuss the presence of a particular group of users in a design process with which they have no direct physical involvement? Here, a key tenet of ANT is that it provides no presupposed theoretical definition of such a presence, as this is exactly what is studied by tracing the relations between actors within the network [41]. In other words, ANT allows for an analysis focused on precisely how physically disabled users are constructed as an actor in the network [42]. Indeed, by drawing inspiration from the theory that attempts to explain the presence of physically disabled users in architectural design processes, "supposedly neutral" design often privileges a particular average understanding of the body [43]. Furthermore, as Yaneva points out, the work of design and architecture is of societal importance, in that it constructs particular realities of which other variations are possible [36]. Based on the above sections, this ties the presence of users in the design process to the agency, affordances, and the attachments of devices. For the study to describe the presence of users with physical disabilities, this makes following the architects as they construct and make use of different types of devices during the design process important.

**4. Analysis**

To identify the presence of users with physical disabilities within the design process of the NES concourse, the empirical data went through three analytical steps. These were (1) scouring field notes and recorded conversations for relevant mentions regarding users and their presence within the project. This was accomplished while (2) looking through the catalogued devices, to locate devices that were mentioned during such conversations and to identify devices through which users with a disability were implicitly or explicitly present. Finally, these two steps were combined in (3) a timeline to draw links between field notes, devices, and the research notes made during steps (1) and (2). This visualization of the design process, as perceived through the lens of a particular academic research question, granted significant analytical clarity. Just as importantly, it allowed the preliminary findings and direction of the analysis to become more accessible for feedback from practitioners at GPA and researchers at the Royal Danish Academy. For illustrative purposes, Figure 2 shows a segment of this final step in the analysis.

The third step of the analysis segmented the field notes into two categories. This categorization was based on steps 1 and 2 of the analysis process, which showed that users with a disability, while never physically involved with the design process, did have an explicit presence at GPA. This presence was primarily related to the multiple rules and regulations surrounding Danish and European accessibility legislation about public transportation. To explore how a group of users could have such a particular presence within the design process, step 3 of the analysis process split the *Y*-axis into two rows. This visually coded the field notes according to whether users with physical disabilities were (A) implicitly or explicitly referred to in conversations or present in devices without referencing obligatory rules or regulation and (B) whether such a presence was tied to the mediation and/or implementation of rules or regulations within the design. For ease of reference, the following analysis will refer to these as the (A) *generalized presence* and (B) *required presence* of users with physical disabilities in the design process of the NES concourse.

*4.1. Generalized Presence*

During the fieldwork, several different types of devices were created by GPA and registered in the catalogue. These included, but were not limited to, plan drawings, section drawings, flow models, physical models, CAD renderings, axonometric projections, and hand-drawn sketches. In their construction and application throughout the design process, these devices served several different purposes. This meant some devices were, e.g., consciously constructed only to communicate spatial solutions for external collaborators

or to explore ideas internally at GPA. Furthermore, some devices saw several iterations or re-emerged at later stages in the design process, whereas others only saw one iteration. Fully describing and tracing the emergence, iterations, and agency of the entire design process is beyond this study, as a specific agential cut has been introduced in the form of structuring a research question. In accordance with this analytically imposed limitation to how the networked design process is described, one interesting purpose of the devices deployed at GPA was their attempts at studying how users in general might experience and use the concourse space. This is the reason for referring to this as *generalized presence*; in such circumstances, users with a disability were present within the design process, to the degree to which their needs overlapped with those qualities generally sought after when designing the NES concourse. This *generalized presence* was the most common of the two types of presence that this study found users with physical disabilities to have during the design process. Moreover, the *generalized presence* was primarily implicit devices and conversations that seldomly led to explicit references to users with physical disabilities.

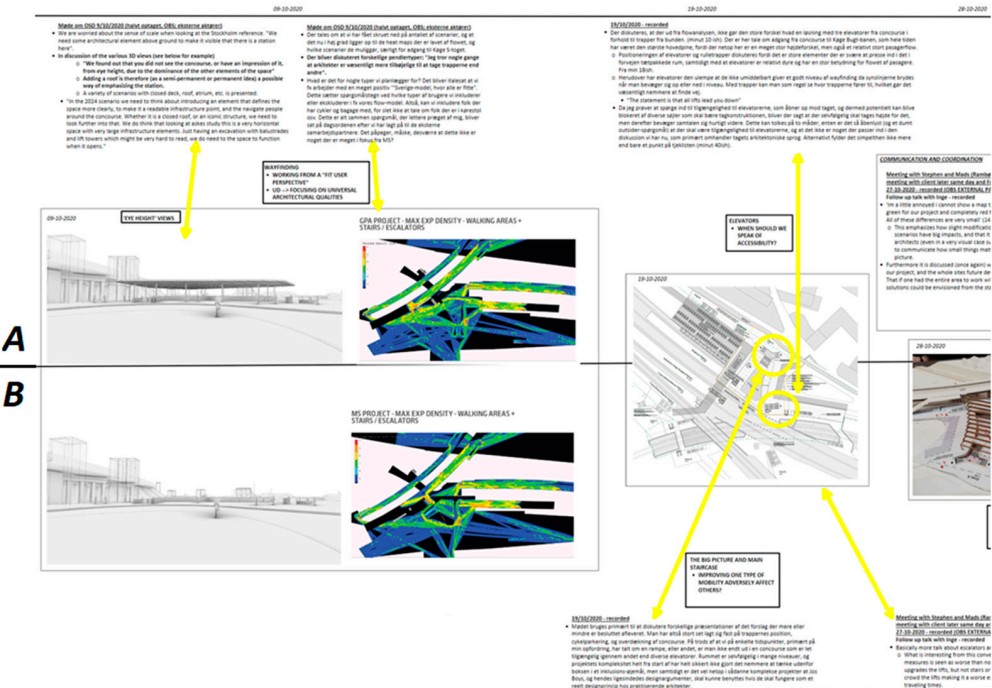

**Figure 2.** Step 3 of the analysis, as visualized through a timeline. A indicates field notes associated with generalized presence. B indicates field notes associated with required presence.

An example of this, from a group of devices created early in the "conceptual design" phase, is presented in Figure 3. The priority of these devices was to understand the position and shape of the escalators, elevators, and staircases that needed to provide access to and through the concourse area. As exemplified by Figure 3, these devices were particularly iterative, as several positions and shapes were tried out. With this intention in mind, several simple "eye-height" 3D perspectives were created (see the bottom two images in Figure 3) to study the implications of different staircases and a potential roof for the concourse. In this way, some devices were constructed to explicitly provide the architects with the average height of a user's perspective of the concourse from the surrounding area. These devices were then presented and discussed amongst the team working on the project and went on to influence how the architects perceived the space. In one conversation, this led one architect to conclude; "we found out that you did not see the concourse, or have an impression of it, from eye height, due to the dominance of the other elements of the space". In turn, this went on to influence the effort put into making the underground concourse visibly distinct at street level from a distance.

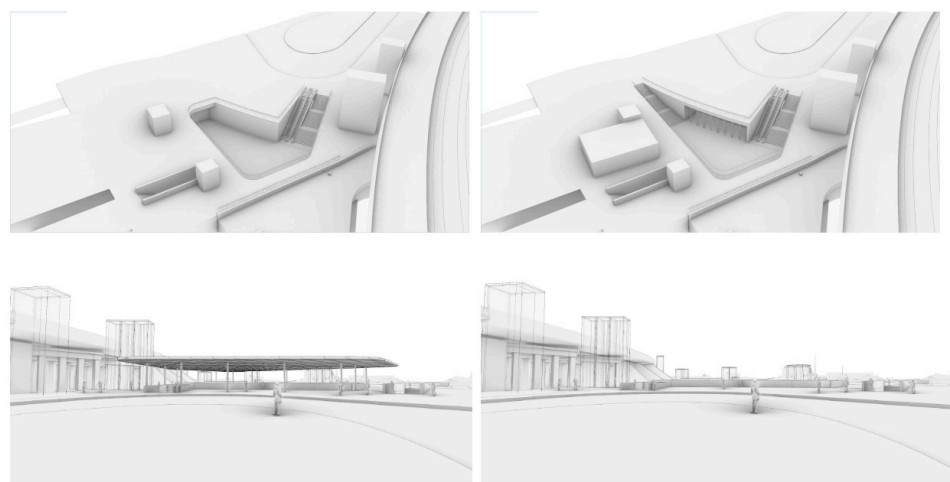

**Figure 3.** Geometrical studies of the NES concourse.

Tightly connected to the knowledge generated by the studies exemplified in Figure 3, the design process saw the construction of several flow studies. The intention of the flow studies was to understand how users would navigate the space and to identify potential bottlenecks. This was done using software that standardized all users to fit within a few set characteristics such as travel speed and a square metre requirement per user. This resulted in the heat maps shown in Figure 4, where green indicates unimpeded commuter traffic and red indicates bottleneck areas where people would feel squeezed together. The advantage of such flow models was that they quickly provided very legible indications of potential flaws in the geometrical layout of NES, which also made them essential to the client at project meetings. In the flow models, green indicates areas where there are no issues with the flow, and red indicates increased congestion. In Figure 4, the image on the left is a flow study of GPA's revision of the design for NES, while the image on the right is a flow study of an older version of the design presented by the client. The slight differences in these two images made one architect comment: "I'm a little annoyed I cannot show a map that is completely green for our project ( . . . ). All of these differences are very small". Due to the simplification of the space that the flow models achieved, there was a reluctance amongst the architects at GPA to rely solely on the information provided by the flow models. Instead, a new flow model was often constructed to test out new geometrical layouts, before returning to devices more akin to those of Figure 3. This meant that flow models were used as a communicatory tool to convince the client of the quality of the design decisions made by GPA and to stress test design decisions based on the use of other types of devices.

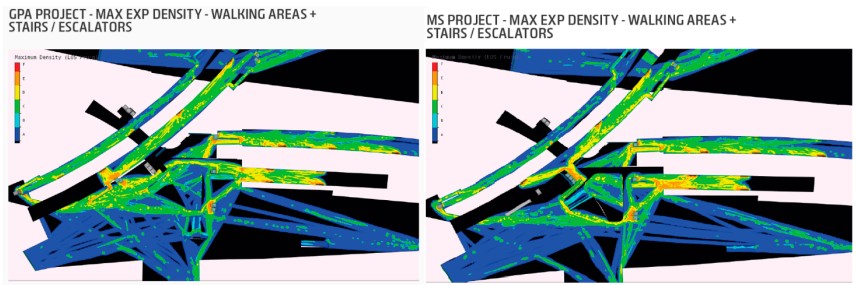

**Figure 4.** Flow models predicting experienced density for pedestrians. Left is GPA's design proposal, and right is the preliminary design proposal by the client.

Another reason for the reluctance that the architects showed about the flow models was the way the software homogenized users to be completely identical. At a separate

instance, while gathered around a table discussing these flow models and how they worked, one architect remarked on the unnuanced nature of a device based on the unlikely scenario of a future "where everybody is fit". The flow studies, despite their importance within the design process, were, therefore, not expected by the architects to adequately represent the diversity of the users that would make use of the future NES concourse. A critique such as this emphasizes that user diversity was an issue that the design process at GPA took seriously; however, it did not spur the architects to adjust the software to represent people needing more space or moving at a slower pace. Despite this, conversations surrounding the flow models showed that the concourse's ability to cater to a diverse user group was important for the architectural quality that GPA sought to deliver. Nonetheless, it was only during discussions tied to flow models that the architects took issue with the diversity of user representation.

As the final example of this section, Figure 5 shows two of the many renderings produced during the "detailed design" phase. Here, the lighting of the future concourse is studied in relation to the possibility of cladding the wall and floor of the concourse with red bricks. These renderings were constructed because it is important for information to be well-lit and accessible, while lighting also plays a big role in how the wayfinding of the concourse will function at night and during the dark Danish winter months. One of the ways in which GPA went about ensuring this was to study the lighting conditions of the concourse at different times of day, with the intent being that information boards should be placed in a well-lit section of the concourse not blocking the main flow of commuters. To this end, devices such as Figure 5 were produced to emphasize how the lighting of the concourse might be handled. In the conversations that occurred surrounding such devices, the spatial experience of the station's future users was central. In a conversation about the lighting of the concourse, one architect referred to the ability to quickly identify routes in and out of the concourse by stating "you feel safe because you can read it". What is interesting from this conversation is that the discussions referred implicitly and explicitly to achieving wayfinding and good lighting as markers of *generalized architectural* quality. Therefore, these qualities were not consciously ascribed to benefit any one type of user more or less than any other types of users.

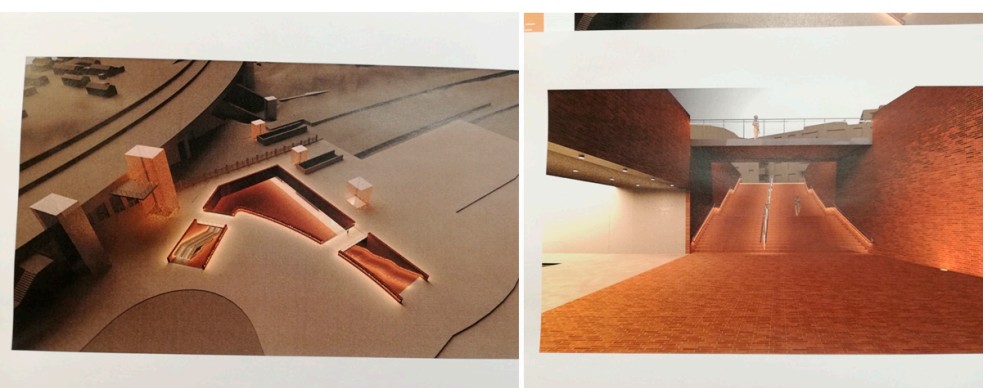

**Figure 5.** Renderings showing evening light conditions of the concourse.

This exemplifies how architects at GPA would construct devices and let themselves be persuaded about the benefits or disadvantages of certain design decisions based on these devices. Moreover, it is an example of the *generalized presence* of users with physical disabilities within the design process, as the devices and conversations in most circumstances defaulted to conceptualizing and depicting "average" able-bodied users. It shows how users with physical disabilities were rarely given an explicit presence within devices and never had devices constructed based on their perceptions of space. What is interesting, for the purposes of this study, is that this influences the affordances of the architectural devices, as they are specifically based on the perspective of an "average" and able-bodied user.

In summary, this indicates that the *generalized presence* of users with physical disabilities was tied to the overall architectural qualities of the project. While this *generalized presence* remained primarily implicit within the design process, the architects at GPA did, at times, show an awareness of differences in bodily fitness and explicitly tied such "diversity" to the quality of the final design. However, this did not translate into the construction of devices that afforded an explicit representation of users with physical disabilities.

### 4.2. Required Presence

Devices that explicitly sought to represent users with physical disabilities and afford spatial analysis founded on this user groups' particular needs were created and circulated during the design process. However, such devices were based on making sure the design adhered to rules and regulations concerning accessibility. *Required presence* refers to those situations in the design process where architects sought to understand and integrate various regulatory demands for accessibility within the design of the concourse. Rules and regulations regarding accessibility were an important topic during meetings, alongside numerous other topics that also needed to be solved during the design process. Indeed, users in wheelchairs and users with visual impairments were mentioned regularly during the design process, in relation to what demands various requirements enforced on the design of the concourse. An example of how rules and regulations regarding accessibility for users with physical disabilities impacted the design process can be observed within the design of the concourse floor.

The placement of the concourse five metres below street level and the roofless dug out space of the concourse mean that the floor of the NES concourse would be observable by commuters and passersby from an elevated position. In addition to being the single largest surface that the architects at GPA were expected to design, this meant the floor of the NES concourse was emphasized as an element of significant aesthetic and architectural importance. This led to the construction of several iterations of devices meant to visualize and test different designs for the floor. At the same time, the design of the floor needed to incorporate tactile guidelines between all entrances and exits within the concourse. This led to the creation of devices that integrated tactile guidelines within the design of the floor in such a way that these guidelines also provided meaningful wayfinding for users with visual impairments. Figure 6 shows one of the plan drawings in which such guidelines were included. Here, a guideline is depicted with a grey line running orthogonally to the stairs and elevators, leading to the station's various platforms on the right-hand side of the concourse. This allowed the architects at GPA to analyse how the guideline might influence the look of the concourse floor, while also attempting to place tactile guidelines to form logical and direct connections between the platforms and the various entrances to the concourse.

Another example of how accessibility requirements enter the design process through architectural devices can be observed in the "material studies" carried out at GPA. Here, Figure 7 shows part of a document put together during the "preliminary design" phase. The top part of Figure 7 shows different wall claddings from other Copenhagen metro stations, and the bottom page shows various types of tactile and visual guidelines from other Danish stations and metro stations. The purpose of this document was to evaluate the aesthetic value, ease of maintenance, cost, and general appropriateness of the materials that would clad the NES concourse's surfaces. Furthermore, as a set of references, the document was constructed to gather what materials were already in use at other Danish train and metro stations in one place. The addition of European train lines to NES meant that the station had to adhere to both Danish and European accessibility regulations in addition to the rules set up by the Danish State Railways (DSB). In turn, this necessitated the introduction of various tactile guidelines and contrast colours to the concourse, which were, therefore, also included as a material to be studied and referenced in the document. This device afforded the evaluation of the many different materials to be used in the future

NES concourse, including those with various requirements that mandated being introduced into the design.

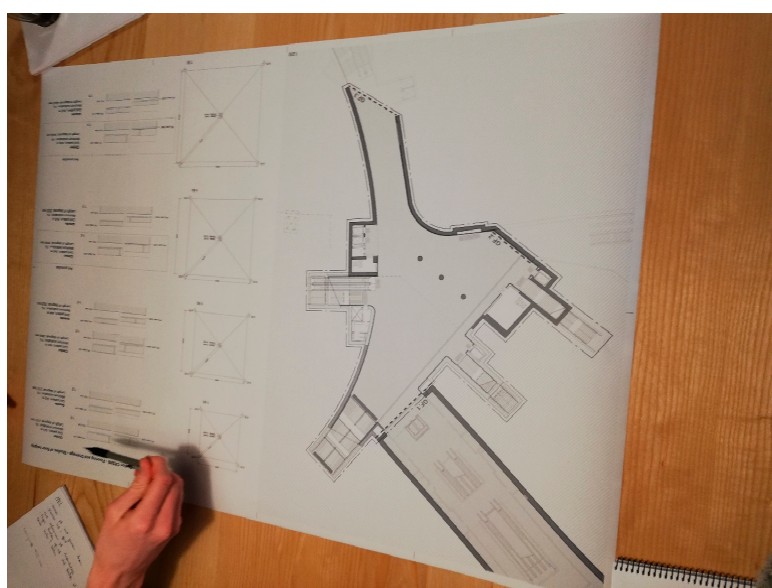

**Figure 6.** Floor plan of the NES concourse with tactile guidelines.

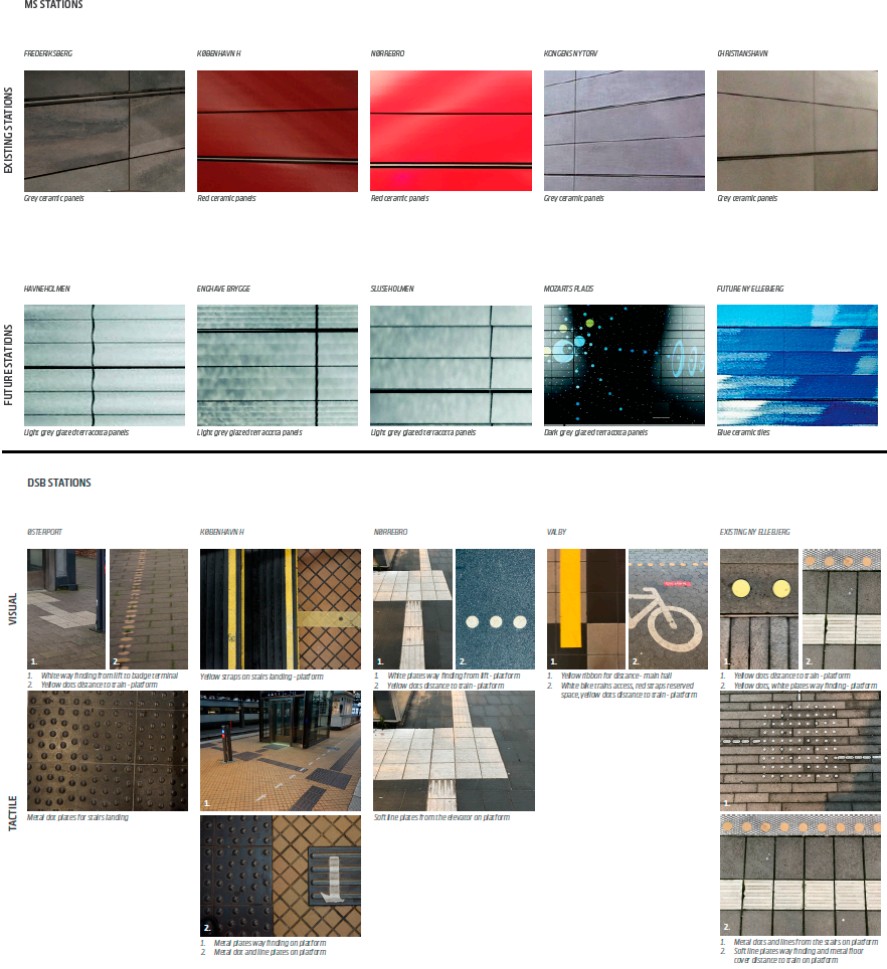

**Figure 7.** Material studies of existing visual and tactile guidelines at other Danish stations and metro stations.

These two examples show how regulations were introduced and analysed within the design process, and they exemplify the *required presence* of users with physical disabilities within the design process. In such instances, the purpose of devices was centred around mitigating the design challenges posed by the mandatory introduction of, e.g., tactile guidelines for the future concourse space. In the discussions surrounding such devices, users with physical disabilities gained an explicit presence in the conversation, though always because of, or in response too, specific spatial requirements imposed on the design. This, on the one hand, meant that users with physical disabilities were explicitly and extensively discussed at some occasions, but, on the other hand, such discussions were always founded in how to most easily adapt requirements into the proposed design for the NES concourse.

## 5. Conclusions and Discussion

By presenting an analysis that attempts to symmetrically describe how architects construct and make use of various visual devices, this study attempts to alleviate the elusiveness that might exist between social scientists and architects about the presence of users with physical disabilities in architectural design processes.

The design process of Ny Ellebjerg Station by Gottlieb Paludan Architects shows several examples of how users with physical disabilities are present through visual devices in an architectural design process. Analytically, these examples have been gathered into those representing a *generalized presence* and those representing a *required presence* by users with physical disabilities. This study finds that while users with physical disabilities did have an explicit *required presence* within the design process, this was tied to the design of the NES concourse needing to fulfill various rules and regulations. On the other hand, an implicit *generalized presence* by users with physical disabilities can also be identified within GPA's overall efforts to achieve architectural quality. As a finding, this is important because it provides a pragmatic and empirically founded case study of how users with physical disabilities, when not integrated through user involvement or similar activities, gain a presence within architectural design processes. This emphasizes the limitation and specificity with which such users become present in architectural design processes through visual devices.

Referencing this study's earlier discussions about the importance of devices and their agency in the architectural design process, this raises questions about how architects perceive the spatial experiences and needs of users with physical disabilities. Specifically, the degree to which devices reproducing the *required presence* of users with physical disabilities end up also reproducing a perception of such users as "others" should be questioned. As the examples from this study have shown, this *required presence* leads to discussions about the maintenance of architectural quality despite rules and regulations. In other words, it becomes a job of adhering to requirements with as few concessions as possible for the architectural vision [44]. If this is the case, a normate critique of architectural design processes is both warranted ethically and empirically. In the case of the design process of the NES concourse at GPA, such normate critiques are met when the *generalized presence* challenges the dominance of the *required presence*. One instance of this was presented in relation to the flow studies of Figure 4—but this is not a frequent occurrence, and it seldomly comes up explicitly in conversations.

### *Implications and Limitations*

It is thought-provoking that the diversity of human bodily abilities is not explicitly present in the spatial analysis afforded by devices in architectural design processes. This is especially true for large public infrastructure projects such as NES, which are supposed to be to the benefit of all citizens of and visitors to Copenhagen. This perhaps indicates that architects need to include new types of devices within their networked design processes, in order to grant a meaningful presence to users with physical disabilities. Such an argument is a return to the normate critique of design practices that was previously described within

this paper. It is possible to indicate that such new devices should be able to explicitly tie accessibility to other matters of architectural quality. This could help architects become more acutely aware of when and how they inadvertently produce normate designs. At the same time, this could train architects to perceive the spaces that they design from the perspective of human beings with significantly different bodily experiences than themselves. Engaging with how such devices might be constructed and integrated into architectural design practices provides an important avenue for future cross-disciplinary pragmatic and empirically founded research.

What such devices might look like is, however, beyond the analysis of this study. Such devices would need to be tested through a methodological and theoretical framework different from ANT. The limitation of conducting pragmatic research of architectural design process is that studies such as this quickly lose their validity and empirical footing, if they attempt to go beyond a descriptive and deconstructivist analysis.

This study can, therefore, describe how the *generalized* and *required presence* of users with physical disabilities creates a gap in such users' representation in architectural design processes. This study cannot conclude what the architectural devices that might fill this gap could look like.

**Author Contributions:** Conceptualization, M.T.M.; methodology, M.T.M.; formal analysis, M.T.M.; investigation, M.T.M.; resources, M.T.M.; data curation, M.T.M.; writing—original draft, M.T.M.; writing—review and editing, M.T.M., M.K. and J.M.K.; visualization, M.T.M.; supervision, M.K. and J.M.K.; project administration, M.T.M.; funding acquisition, M.T.M., M.K. and J.M.K. All authors have read and agreed to the published version of the manuscript.

**Funding:** This research was funded by the Bevica Foundation and the Innovation Fund Denmark (0153-00044B).

**Informed Consent Statement:** All architects participating in the NES project at Gottlieb Paludan Architects, as well as external stakeholders present at meetings, were made aware of the intentions and presence of the researcher. Informed consent was also signed and collected from all architects participating in the NES project at Gottlieb Paludan Architects. Furthermore, an email was sent out by Gottlieb Paludan Architects to the external partners in the NES project, including the client and project manager organizations, describing the intentions of the research project and the type of data that would be collected.

**Data Availability Statement:** Due to the sensitive nature of the data of this project, which include both company secrets and potentially infringing GDPR information about individuals, the data are only available through specific requests.

**Acknowledgments:** We would like to acknowledge the Bevica Foundation for providing the funding as well as Gottlieb Paludan Architects for granting the access to their office that made this research possible.

**Conflicts of Interest:** The authors declare no conflict of interest.

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
