# Peer review of "Elusive Users: The Presence of Physically Disabled Users within Architectural Design Processes"

_2673-8945, doi:10.3390/architecture3010003_

Round 1
Reviewer 1 Report
The study is scientifically rigorous. The method, subject and material discussed are suitable for the journal. Within the scope of the study, the ideas were clearly expressed. The author has cited current and relevant studies. Authors can separate the methodology section where they describe the method and steps of the research. Authors can address the evaluations of their analysis under the heading of findings and evaluation.
Author Response
Dear reviewer
Thank you for your feedback. We have responded to your following points:
- Authors can separate the methodology section where they describe the method and steps of the research:
- We appreciate the comment that the structure of the paper could be different. We have elected to keep the structure of the paper as is as we believe it is important to gain an understanding of how the analysis took shape immediately before the analysis is presented, in order to better follow and understand the arguments presented in the analysis.
- Authors can address the evaluations of their analysis under the heading of findings and evaluation:
- The text has added a ‘implications and limitations’ section which attempts to evaluate more explicitly the findings of the analysis in addition with positioning these implications in relation to the limitations of this study.
Reviewer 2 Report
Interesting paper with a sought-after focus on design process. Here are my comments:
State of the art (line 34) could be extended
You could describe the design process and the actors involved in it more thoroughly
You could develop your thoughts on the need to include new types of devices (line 468)
You could acknowledge the limitations of your research
You could be harsher in your conclusions to make your point clear
Author Response
Dear reviewer
Thank you for your well structured and constructive feedback. We have responded to your following points:
- State of the art (line 34) could be extended:
- The text now includes an extended argument for why research on involvement of physically disabled users in architectural processes differs from the emphasis of this paper, while also providing additional references.
- You could describe the design process and the actors involved in it more thoroughly:
- The text now includes an extended description (from line 124) of the surrounding context and actors involved within the design process. Due to the complexity of the case project, and the specificity of the argument which the study is attempting to reach, it is believed that adding more than this would primarily serve to add unnecessary information and render the text more impenetrable.
- You could develop your thoughts on the need to include new types of devices (line 468):
- The text now includes a new section in the conclusion & discussion section which expands upon the idea of introducing new devices to architecture. This section also explains why the study is limited in its ability to comment on what such devices might look like. Furthermore the final segment expands the criticism levelled at architectural design processes slightly. Going beyond this criticism is not in line with the theoretical and analytical intentions of the study, nor can such criticism be empirically supported by the presented analysis.
- You could acknowledge the limitations of your research:
- See above comment
- You could be harsher in your conclusions to make your point clear:
- See above comment